# Synthesis, Characterization, and Optimization Studies of Polycaprolactone/Polylactic Acid/Titanium Dioxide Nanoparticle/Orange Essential Oil Membranes for Biomedical Applications

**DOI:** 10.3390/polym15010135

**Published:** 2022-12-28

**Authors:** Jorge Ivan Castro, Stiven Astudillo, Jose Herminsul Mina Hernandez, Marcela Saavedra, Paula A. Zapata, Carlos Humberto Valencia-Llano, Manuel N. Chaur, Carlos David Grande-Tovar

**Affiliations:** 1Grupo de Investigación SIMERQO, Departamento de Química, Universidad del Valle, Calle 13 No. 100-00, Santiago de Cali 76001, Colombia; jorge.castro@correounivalle.edu.co (J.I.C.); manuel.chaur@correounivalle.edu.co (M.N.C.); 2Grupo de Materiales Compuestos, Escuela de Ingeniería de Materiales, Facultad de Ingeniería, Universidad del Valle, Calle 13 No. 100-00, Santiago de Cali 760032, Colombia; joan.astudillo@correounivalle.edu.co (S.A.); jose.mina@correounivalle.edu.co (J.H.M.H.); 3Grupo de Polímeros, Facultad de Química y Biología, Universidad de Santiago de Chile, USACH, Santiago 9170020, Chile; alecram.saavedra@gmail.com (M.S.); paula.zapata@usach.cl (P.A.Z.); 4Grupo Biomateriales Dentales, Escuela de Odontología, Universidad del Valle, Calle 4B # 36-00, Cali 76001, Colombia; carlos.humberto.valencia@correounivalle.edu.co; 5Grupo de Investigación de Fotoquímica y Fotobiología, Facultad de Ciencias, Universidad del Atlántico, Carrera 30 Número 8-49, Puerto Colombia 081008, Colombia

**Keywords:** biocompatibility, connective tissue, orange essential oil, polycaprolactone, polylactic acid, titanium dioxide nanoparticles

## Abstract

The development of scaffolds for cell regeneration has increased because they must have adequate biocompatibility and mechanical properties to be applied in tissue engineering. In this sense, incorporating nanofillers or essential oils has allowed new architectures to promote cell proliferation and regeneration of new tissue. With this goal, we prepared four membranes based on polylactic acid (PLA), polycaprolactone (PCL), titanium dioxide nanoparticles (TiO_2_-NPs), and orange essential oil (OEO) by the drop-casting method. The preparation of TiO_2_-NPs followed the sol–gel process with spherical morphology and an average size of 13.39 nm ± 2.28 nm. The results show how the TiO_2_-NP properties predominate over the crystallization processes, reflected in the decreasing crystallinity percentage from 5.2% to 0.6% in the membranes. On the other hand, when OEO and TiO_2_-NPs are introduced into a membrane, they act synergistically due to the inclusion of highly conjugated thermostable molecules and the thermal properties of TiO_2_-NPs. Finally, incorporating OEO and TiO_2_-NPs promotes tissue regeneration due to the decrease in inflammatory infiltrate and the appearance of connective tissue. These results demonstrate the great potential for biomedical applications of the membranes prepared.

## 1. Introduction

Several diseases, congenital disorders, and injuries are the leading causes of human tissue malfunction forcing tissue substitution through surgical procedures. The tissues’ environment involves an extracellular matrix consisting of fibrillar proteins, proteoglycans, glycosaminoglycans (GAGs), and minerals. Currently, artificial scaffolds are used as support structures for cell culture and growth to repair damaged tissues or organs with the primary objective of temporarily assisting cell regeneration, as this support is resorbed, either during the healing process or afterward, leading to the generation of new tissue with the desired properties [1].

Different authors have focused on developing materials that combine the benefits of each component, taking into account the specific biological, clinical, and medical aspects of the tissue defect. In this sense, using polymeric matrices comprising a mixture of one or more polymers (natural or synthetic) becomes an alternative to the mechanical, thermal, or biological challenges they may exhibit when used for scaffolding [2].

Polylactic acid (PLA) is a semicrystalline polymer obtained from the polymerization of lactic acid, a by-product of starch fermentation. PLA presents excellent tensile strength, broadly used in several fields due to its wide range of applications. However, PLA brittleness limits some applications [3,4].

Polycaprolactone (PCL) is a thermoplastic that can be obtained by polymerizing ε-caprolactone using a catalyst involving a ring-opening mechanism, also exhibiting a high deformation. PCL is moldable and extrudable, and can be homogeneously mixed with several polymers [5]. Nevertheless, PLA and PCL have been considered critical polymeric matrices because they do not induce a high level of bioactivity, since tissue responses have been described with a low level of inflammation and low vascularization rates [6,7]. Additionally, PCL shows superior rheological and viscoelastic behavior compared to other aliphatic polyesters, which makes it easily processable in the fabrication of biodegradable devices [8,9]. There are different ways to solve the individual problems presented by these polymers; for instance, the addition of nanofillers [10].

In the last decade, inorganic ceramic nanomaterials, such as titanium dioxide (TiO_2_), have been necessary for several applications, such as photocatalysis, sensory systems, and the reinforcement of biomaterials [11]. TiO_2_ nanoparticles’ biocompatibility and anti-corrosive properties make them attractive fillers within biodegradable polymeric matrices [12]. Recently, titanium dioxide nanoparticles with biopolymers and essential oils have been studied to enhance mechanical, thermal, and biodegradable properties and stimulate antibacterial and regenerative activity. TiO_2_ provides bioactive properties, while the biopolymers provide chemical properties [13,14].

On the other hand, essential oils are volatile active compounds that improve structures and mechanical properties in polymeric matrices; they also provide antimicrobial properties related to the organic structural component [15]. Orange essential oil (OEO) belongs to the citrus genus. It is a complex combination of volatile components, such as non-oxygenated and oxygenated terpenes. D-limonene (LN), (1-methyl-4-(1-methyl phenyl) cyclohexane) is one of the main components of citrus essential oils (25%–98%), depending on the species and varieties of citrus [16]. However, due to its volatility and sensitivity to deterioration, it may lose its function and cannot be applied as a long-term coating [17]. Nevertheless, studies have been conducted that have allowed the encapsulation of OEO in a gelatin matrix and in nanofibers of tannic acid crosslinked with gelatin, where crosslinking between the tannic acid and gelatin decreased the encapsulation capacity of the essential oil, releasing the oil in a controlled manner [17]. Thus, many research groups have included nanofillers and natural ingredients, such as essential oils, in polymeric matrices, to search for novel active properties [18].

In this sense, nanoparticles such as ZnO-NPs, SiO_2_-NPs_,_ and TiO_2_-NPs influence the PLA aging process. The authors realized that the nature and properties of these scaffolds change according to the nanoparticles added [19]. On the other hand, adding ZnO altered the crystallinity in PLA-based fibers affecting photodegradation behavior [19]. However, despite the study in different fields of PLA nanocomposites with nanoparticles, there is a need to improve the stability and biocompatibility under physiological conditions to prolong the durability of nanocomposites with physical mixtures containing a PCL/PLA polymeric matrix as well as the orange essential oil. There is still a great need to determine whether blending nanofillers with other components improves biocompatibility, especially in vivo studies. For this reason, in this work, preliminary biocompatibility was evaluated through subdermal implantations of four PCL/PLA/TiO_2_-NPs/OEO-based formulations to improve biocompatibility to envision the potential for biomedical applications.

## 2. Materials and Methods

### 2.1. Materials

The chemicals used in the investigation were of analytical degree and were used without purification unless otherwise stated. The PCL used was a 6800-layer provided by Perstorp Company (Warrington, UK), and contained a molecular mass of 80,000 g/mol, with a flow rate of 3 g/10 min. PLA (<2% D-isomer, 200,000 Da) was purchased from NatureWorks (Minnetonka, MI, USA). For the generation of TiO_2_ nanoparticles, 99% titanium isopropoxide (TTIP), 2-propanol, and nitric acid provided by Aldrich (Palo Alto, CA, USA) were used as precursors. Finally, OEO was provided by Mamys (Madrid, Spain), and the relative chemical composition used gas chromatography coupled with mass spectrometry (GC-MS).

### 2.2. Synthesis of TiO_2_ Nanoparticles

The synthesis of TiO_2_ nanoparticles was performed based on the methodology previously reported [20]. Briefly, we prepared two different solutions: the first consisted of 15 mL of TTIP mixed with 15 mL of 2-propanol (solution 1). In contrast, the second consisted of 250 mL of distilled water and a solution of HNO_3_ 3M maintaining the pH at 2.0. Subsequently, solution one was added dropwise over the solution at pH 2.0 with constant stirring; once solution one was added, the final solution remained at 60 °C for 20 h. Finally, the solvent was evaporated, precipitating yellow crystals, which were washed with ethanol to obtain a white–yellow powder. This solid was subjected to calcination at 400 °C for two h.

#### Characterization of Nanoparticles of TiO_2_

TiO_2_ NP characterization was reported elsewhere [20]. The study of the TiO_2_ NP morphology and particle size used transmission electron microscopy (TEM) in a JEOL ARM 200 F (Tokyo, Japan). The characterization added a drop of TiO_2_ on a standard carbon-coated copper grating (40 mesh) with solvent evaporation. For the calculation of the particle size, we averaged the size of 100 nanoparticles using image J software. On the other hand, the characteristic diffraction planes for the nanoparticles were obtained through a PANalytical X0Pert PRO diffractometer (Malvern Panalytical, Jarman Way, Royston, UK) using Cu Kα1 (1.540598 Å) and Kα2 (1.544426 Å) radiation, with a 45 kV voltage accelerator in a range 2q between 5 and 80°. FTIR experiments on TiO_2_ used attenuated total reflectance (ATR) mode and a diamond tip accessory (Shimadzu, Kyoto, Japan).

### 2.3. OEO Composition

OEO composition used gas chromatography coupled to mass spectrometry (GC-MS) with an AT6890 series plus gas chromatograph (Agilent Technologies, Palo Alto, CA, USA) and a mass selective detector (Agilent Technologies, MSD 5975) in full scan mode. An apolar C6-C25 hydrocarbon column was the standard reference. The column used: DB-5MS (J & W Scientific, Folsom, CA, USA), 5% -Ph-PDMS, 60 m × 0.25 mm × 0.25 mm, Split mode (30:1) injection, Viny = 2 mL. The identification of components used the component’s retention index (RI) for relative identification in the Adams database (Wiley, 138 and NIST05, for Agilent, Santa Clara, California, CA, USA).

### 2.4. Preparation of the Membranes of PCL/PLA/TiO_2_/OEO

Membrane preparation followed a previous report [21] using the weight ratios reported in Table 1. Each component was dissolved in chloroform and mixed according to Table 1. The one-hour ultrasonic bath treatment (Branson, Madrid, Spain) removed the gas, giving a final concentration of 4 wt.%.

The homogeneous solution was placed on glass molds for 24 h and left to dry at room temperature. Then, these formulations were cured in a preheated oven at 40 ± 0.2 °C to obtain the PCL/PLA/TiO_2_/OEO membranes.

#### Characterization of PCL/PLA/TiO_2_/OEO Membranes


*Fourier Transform Infrared Spectroscopy (FTIR)*


PCL/PLA/TiO_2_/OEO membranes functional groups determination used an affinity-1 IR spectrophotometer (Shimadzu, Kyoto, Japan) between 500 and 4000 cm^−1^ in transmittance mode and a diamond tip accessory (Shimadzu, Kyoto, Japan).


*X-ray Diffraction*


The corresponding diffraction planes for each formulation were elucidated using the same conditions and equipment described for TiO_2_ NPs in a 2θ range between 5–40°.


*Scanning Electron Microscopy (SEM)*


Membrane surface morphology was analyzed with a scanning electron microscope (SEM) (Hitachi TM 3000, Musashino, Tokyo, Japan) in secondary electron mode and a voltage acceleration of 20 kV. Samples were coated with gold for better electron conductivity.


*Thermal Analysis*


Thermogravimetric analysis (TGA) used a NETZSCH TG 209 F1 Libra (Mettler Toledo, Schwerzenbach, Switzerland) equipment, with a 10 °C/min between 25–800 °C under nitrogen atmosphere (50 mL/min).

Glass transition temperature (T_g_), melting temperature (T_m_), crystallization temperature (T_cc_), enthalpy of fusion (
$\Delta H_{m}$
) and enthalpy of crystallization (
$\Delta H_{CC}$
) were determined by differential scanning calorimetry (DSC) using a DSC1/500 (Mettler Toledo, Schwerzenbach, Switzerland) equipment. Membranes were heated from −25 °C to 250 °C, with a heating rate of 10 °C/min and a nitrogen flow of 60 mL/min. All the DSC data were subtracted from the second heating sweep to delete the thermal polymer memory. The TGA and DSC data were analyzed through TA instruments Universal Analysis Software 2000 version 4.5A.

The crystallinity (%) calculations used the enthalpy of fusion (*ΔH°_m_*) of PLA (100% crystalline) according to Equation (1) [22].

(1)
$$X_{c}=\frac{\left(\Delta H_{m}-\Delta H_{CC}\right)}{\Delta H_{m}^{{}^{\circ}}\left(1-x\right)}$$

where 
$\Delta H_{m}^{{}^{\circ}}$
 s 93 corresponds to the enthalpy of fusion of the ideal PLA at 100% crystallinity [23]. 
$\Delta H_{m}$
 and 
$\Delta H_{CC}$
 are the enthalpy of fusion and enthalpy of crystallization of the nanocomposites in J/g, and the wt.% of PLA is given by 1 − x term.

### 2.5. Preliminary Biocompatibility Analysis of In Vivo Membranes

#### 2.5.1. Surgical Preparations of Biomodels

The subdermal surgical implantation of biomodels followed the standard UNE: 10993-6 (Biological evaluation of medical devices—Part 6: Tests of local effects after implantation. ISO 10993-6: 1994). Nine Wistar rats (*Rattus norvegicus* domestic), four months old with an average weight of 380 g, were implanted with the membranes in the dorsal area for each formulation.

Biomodel sedation was achieved using ketamine 30 mg/kg and xylazine 70 mg/kg (HOLLIDAY SCOTT S.A., Buenos Aires, Argentina) and xylazine 70 mg/kg (Xilaxyn-Virbac., Bogotá, Colombia) intramuscularly. After complete sedation, a procedure of trichotomy was performed on the dorsal surface using an antiseptic (iodine solution). At the same time, the anesthetic for the zone was Lidocaine 2% with Epinephrine 1:80,000 (Newcaina, Guarne, Antioquia, Colombia). The intervention used a Bard-Parker scalpel (blade 15), creating six cuts (10 cm length), and six pockets 3 cm deep were created with a dental peristome. Once the pockets were prepared, the five samples of the experimental material (10 mm × 5 mm × 3 mm membranes) were implanted as a fast-absorbing control material.

#### 2.5.2. Histology Tests

After 90 days of membrane implantation, the biomodels were euthanized using an intraperitoneal application of 150 mg/kg pentobarbital sodium 100–150 mg/kg (Euthanex-INVET, Medellín, Colombia). Sample recovering and fixing was achieved with a buffered formalin for 48 h.

After fixation, the zones of interest were fragmented in equal proportions and dehydrated by immersion, varying the alcohol concentration in ascending order from 70% to 100% *v*/*v*. Finally, inclusion procedures were performed by diaphanization with xylol and infiltration with kerosene using an Auto-technicon Tissue Processor™ (Leica Microsystems, Mannheim, Germany). Histological analysis was performed using kerosene with the Thermo Scientific™ Histoplast Paraffin™ kit (Fisher Scientific, Waltham, MA, USA). Afterward, we cut 5 µm sections using a Leica microtome (Leica Microsystems, Mannheim, Germany) and placed them on glass Petri dishes for 48 h. Subsequently, the samples were stained using Masson’s hematoxylin-eosin and trichrome stains. Finally, we used a Leica optical microscope equipped with an imaging suite (Leica Microsystems, Mannheim, Germany) for the optical image preparation of samples.

This research followed the Universidad del Valle Ethics Committee guides in Cali, Colombia, for animal testing with possible biomedical applications using the resolution CEAS 012-019.3.

## 3. Results and Discussion

### 3.1. Characterization of the Nanoparticles of TiO_2_

TiO_2_ NP preparation followed a known procedure [20]. The size and morphology of the TiO_2_ nanoparticles were analyzed by TEM (Figure 1) with spherical characteristics and an approximate size of 13.39 nm ± 2.28 nm. The TiO_2_ crystallographic study was consistent with an anatase structure due to the angular location and intensity of the peaks obtained at specific 2θ values 25°, 38°, 48°, 54°, 63°, 69°, and 75° corresponding to the (101), (004), (200), (211), (105), (204) and (220) planes for TiO_2_ [24]. The nanoparticles’ FTIR spectrum showed symmetric strain bands at 3060 cm^−1^ and bending bands at 1641 cm^−1^ for -OH due to adsorbed water molecules. The peaks around 821 and 1132 cm^−1^ corresponded to the stretching of the Ti-O-Ti and Ti-O-C bonds, respectively [20].

### 3.2. Composition of OEO

According to the GC-mass analysis of OEO (Appendix A), there were twenty-four compounds. The compounds corresponded to limonene (89.6%) as the primary component, followed by β-myrcene (2.9%), camphene (1.2%), α-Pinene (1.1%), and linalool (0.7%), among others. Our results are consistent with the findings of Galvao and coworkers [25].

### 3.3. Preparation of PLA/PCL/TiO_2_/OEO Membranes

PCL/PLA/TiO_2_/OEO membranes procedure consisted of the *drop-casting* method by combining in situ each of their components to generate four formulations (F1–F4) according to Table 1. Considering the solid covalent and non-covalent interactions that organic polymers can present, PCL and PLA can form a porous matrix for TiO_2_ nanoparticles to anchor directly on the surface. Therefore, different formulations containing different fillers, such as TiO_2_ nanoparticles and OEO, were prepared separately using the same polymeric matrix to observe the effect on the porosity and biocompatibility of the material. Finally, the union of these two components in the same matrix presented a synergistic effect without deteriorating the physicochemical properties.

#### 3.3.1. FT-IR for PLA/PCL/TiO_2_/OEO Membranes

To demonstrate the presence of each component in the formulation, Fourier transform infrared spectroscopy (FT-IR) was used to describe the bands associated with their functional groups and possible interactions (Figure 2). In general, there were symmetric and asymmetric strain bands of the -CH bond between 2945–3001 cm^−1^, the stretching band of the C=O group at 1756 and 1722 cm^−1^ belonging to the ester functional group of PCL and PLA, respectively, and the asymmetric stretching of the aliphatic C-O-C ether group of both PCL and PLA was observed in the range 1182–1296 cm^−1^. Finally, the carbonate chain stretching of both PCL and PLA attributed to the CH_3_ methyl group oscillating band was found between 960–830 cm^−1^.

For the membranes containing TiO_2_ nanoparticles (F2 and F4), the characteristic band at 438 cm^−1^ was absent, probably due to the tiny amount by weight for each formulation [26]. On the other hand, in the films containing OEO, shifts at high wavenumbers were observed, especially of the strain band at 2945 cm^−1^ corresponding to the -CH bond. In addition, the band at 1457 cm^−1^ showed an increase in intensity associated with the elongation vibrations of the C=C bond of the olefinic groups of the essential oil [27].

Nevertheless, it is interesting that for F3 and F4, there was a broadening of the C=O group band (1756 cm^−1^), in addition to the decrease in both the C-O-C bond band (1296 cm^−1^) and the C-H bond intensity (3001 cm^−1^), which evidences the presence of OEO, suggesting hydrogen bonding interactions between the polymers present and the OEO (mainly limonene). These observations are congruent with previous work with similar compositions [28].

#### 3.3.2. XRD of the PLA/PCL/TiO_2_/OEO Membranes

Figure 3 shows the effect on the crystallinity of the PCL/PLA polymer blend upon introducing OEO and TiO_2_. Crystallographic data show the orthorhombic nature of PLA with consistent crystalline planes at 2θ diffraction angles of 14° (011), 17° (101), 19° (012), and 22° (013)). Moreover, it was possible to observe the peaks of PCL at 21° and 24° corresponding to the (110) and (200) planes. The blending behavior between the polymers is consistent with the crystallographic profile of each polymer for all the membranes investigated, demonstrating that PLA and PCL retain the same diffraction planes when blended [29].

On the other hand, the addition of TiO_2_ did not modify the crystallographic profile of the PCL/PLA mixture, which was reported in formulations with similar PCL/PLA/TiO_2_ compositions [13]. It is probable that TiO_2_ nanoparticles disperse on the polymeric matrix, fill the empty spaces, and do not affect the crystallinity of the mixture when used in lower proportions than 5 wt.%, as seen in previous works [10,30]. Similarly, the inclusion of OEO did not influence the crystallographic profile of the PCL/PLA blend because the essential oil spreads throughout the membrane uniformly, and part of the oil evaporates due to its volatile composition, causing porosity. Similar results have been reported previously [31].

#### 3.3.3. Thermal Analysis of PLA/PCL/TiO_2_/OEO Membranes

Thermogravimetric analysis (TGA) was used to determine the thermal properties of the membranes from the percentage loss in weight concerning temperature. Figure 4A shows the different thermograms for formulations F1, F2, F3, and F4. The thermal decomposition for formulation F1 (PCL/PLA) shows three stages of thermal degradation. The first stage corresponds to the weight loss at 309 °C attributed to PCL degradation. Initially, the hydroxyl terminals are detached from the polymer chain, causing depolymerization after the polymer is cleaved by cis elimination [5]. The second stage of degradation has a peak at 358 °C corresponding to intramolecular transesterification from the lactide and some cyclic oligomers, followed by the reaction of PLA fragments resulting from cis-elimination between carbon oxides and acetaldehyde for the formation of acrylic acid [3]. Finally, the third degradation stage has its maximum degradation peak at 403 °C, attributed to residues remaining after cis-elimination that originate detachments from the hydroxyl end of the polymeric chain of PCL.

On the other hand, the PCL/PLA membranes (F1) showed a higher degradation temperature owing to the stability of both polymers. In contrast, membranes with lower PCL contents decreased their degradation temperature (F2–F4) (Figure 4B, DTGA). Additionally, TiO_2_ nanoparticles reduce the degradation temperatures because the TiO_2_ nanoparticles are mainly in the polymer matrix phase catalyzing degradation [13]. Concerning the F3 membrane, which contains OEO within the polymer matrix, there was an increase in thermal stability because the OEO structurally has thermally stable components. Additionally, the loss of broadening of the degradation peaks is possibly due to the lower amount of water in the membrane, resulting from the hydrophobic nature of the main components of the essential oil OEO. Finally, for the F4 membrane, an increase in thermal properties was observed, probably due to the addition of aromatic structured OEO and high thermal specification nanoparticles, such as TiO_2,_ absorbing the applied thermal energy and storing it simultaneously [14,32].

Differential scanning calorimetry (DSC) measurements provided information on the physical properties within the membranes, as observed in Figure 5. In this thermogram, the glass transition temperature T_g_, crystallization temperature T_cc_, melting temperatures T_m1_ and T_m2_ of PLA, and crystallinity percentage X_c_ were determined for each formulation, with comparable results with previous studies [33]. Additionally, the melting temperature from PCL was labeled as Tm_3,_ as shown in Table 2.

For semicrystalline PLA, the curves showed three thermal transitions. An endothermic peak (ca. 54 °C) was associated with structural relaxation during the glass transition. Second, the exothermic transition peaks rose from the cold crystallization process (ca. 98 °C), while two melting temperatures resulted from the crystalline domains. Additionally, it was possible to distinguish the T_m_ from PCL, which was close to the T_g_ of the PLA. Therefore, it was not easy to calculate the crystallinity percentage from PCL polymer.

According to the previous table, there was a decreased glass transition temperature in F4 membranes concerning the other formulations. The essential oil incorporation into the PLA promoted a decrease in the glass transition temperature (T_g_) compared to the neat PLA. These results were similar to those in which the D-limonene had a plasticizer effect that increased PLA chain mobility, with a slight glass transition temperature reduction [34].

According to Equation (1), PLA crystallinity decreased from 5.2% to 0.6% because the addition of TiO_2_ hindered the mobility of the PLA chains, reducing the crystallinity. However, the low-viscosity OEO provided greater mobility for the PLA chains [34]. The two endothermic peaks at 135 and 146 °C (T_m1_ and T_m2_) were attributed to PLA undergoing cold crystallization due to different crystalline phases. The first corresponded to the pseudo-orthorhombic, pseudo-hexagonal and orthorhombic α form, melting at a higher temperature, while the second corresponded to the β form (orthorhombic o trigonal), which melted at a lower temperature. Nowadays, there is a general comprehension that the α- and β- chain helical conformations have the same energy; therefore, they assemble in two different crystal structures [35]. However, a decreased endothermic peak (135 °C) for F3 suggested a transformation from the beta to the alpha crystalline form due to an increased PLA crystallization rate upon heating by OEO [36]. On the other hand, the T_cc_ exothermic peak rose from polymer chain crystallization during the cooling cycle [37].

#### 3.3.4. Scanning Electron Microscopy (SEM) of PLA/PCL/TiO_2_/OEO Membranes

Figure 6 shows the surface micrographs of the PCL/PLA/TiO_2_/OEO membranes. The SEM images showed a porous structure with gaps that could facilitate interactions of new compounds due to the low compatibility between the polymer matrices. Additionally, the small spheres observed in the membranes were due to the spherical-shaped PCL in a continuous PLA matrix. The diameter of these spheres increased with increasing PCL concentration. Furthermore, the dispersion of PCL spheres in the PLA matrix at 20% and 17% for the F3 and F4 formulations was uniform.

Additionally, sample F3 presented dense microholes without the accumulation of TiO_2_ nanoparticles. These micro holes were probably due to the partial evaporation of the volatile components of the essential oil from the OEO during the membrane formation process. The OEO enhances TiO_2_ dispersion within the membrane, which may be related to lower -OH groups on the TiO_2_ surface, causing a decrease in the specific surface area value of TiO_2_ with the presence of OEO [38].

Formulations F2 and F4 with 3 wt.% TiO_2_ presented white particles corresponding to TiO_2_ nanoparticles. Different researchers confirmed that nanoparticles are concentrated in the PLA phase with a small population at the polymeric interface due to the poor compatibility between TiO_2_ nanoparticles and PCL [10,30].

#### 3.3.5. In Vivo Biocompatibility Test of the PLA/PCL/TiO_2_/OEO Membranes

The histological analysis of the membranes recovered after the implantation period in biomodels allowed us to determine the interaction between the membranes with the defective tissue or organ, to fill tissue voids and support adhesion and cell proliferation. Furthermore, during the recovery of the samples, a macroscopic inspection of the implanted areas was performed, considering the UNE-EN ISO 10993-6:2017 standard related to the biological evaluation of products and the study of the local effect after implantation.

Figure 7 corresponds to the dorsal image of one of the biomodels after 30 days of implantation. A similar appearance was observed in the three implantation periods (30, 60, and 90 days), with complete hair recovery in all cases (Figure 7A). When the trichotomy was performed (Figure 7B), the surgical lesions in the implantation area were wholly healed and recovered. In Figure 7C, it is possible to observe four regions intervening in the internal surface of the skin figure. In all cases, complete healing of the areas implanted was observed, without necrosis or pus presence.

Furthermore, the encapsulation of the foreign body (implanted material) allows the reappearance of injured tissues because the fibrous tissue surrounds the material while phagocytic cells remove the material [39].

Figure 8 shows how in the implantation results of F4 at 90 days, the fibrous capsule was no longer observable because the material was almost completely resorbed. The histological analysis of the F1 formulation, 30% PCL:70% PLA, showed practically complete resorption of the material 90 days after implantation. At 30 days in the implantation area, a good part of the material was resorbed with an inflammatory infiltrate. In addition, it was possible to observe that the process occurred in an area surrounded by a capsule of connective tissue (Figure 8: F1A and F1B). As the magnification of the microscope’s objective lens increased, it was observed that this capsule was formed by type I collagen fibers, evidenced by Masson’s Trichrome (MT) technique (Figure 8, F1B). In addition, it was found that in the encapsulated area’s interior, the material portions were separated by type I collagen fibers (Figure 8, F1C). The healing process begins with the fibrous encapsulation of type I collagen surrounded by a lymphocytic inflammatory infiltrate, promoting the resorption of the material through its fractionation.

On the other hand, 60 days after implantation, the presence of an area with inflammatory cells was observed (Figure 8, F1D and F1E), with an abundant presence of type I collagen (Figure 8, F1F); that is, during the implantation process, trauma occurred that disrupted the homeostasis of the skin. Therefore, the skin goes through 4 stages: hemostasis, inflammation, proliferation, and remodeling, allowing it to return to its initial state. Thus, the presence of inflammatory cells also causes reparative cells. However, severe trauma generates altered healing due to the prolongation time of inflammatory cells producing lacerations or ulcers and the presence of pus. After 90 days of implantation, the area with inflammatory infiltrate had decreased significantly (Figure 8, F1G and F1H), but with fragments of the material with type I collagen in the implantation area.

The F2 formulation with a composition of 27% PCL:70% PLA:3% TiO_2_ presented a histological behavior similar to that reported for F1 in the 30 days. In the implantation zone, the material was observed surrounded by a fibrous capsule, fragmented in its interior and with the presence of collagen type I fibers and inflammatory cells, exhibiting a typical foreign body reaction (Figure 8, F2A, F2B, and F2C).

At 60 days, portions of the implanted material in the degradation process by inflammatory cells were still visible (Figure 8, F2D and F2E). Still, at higher magnification, it was possible to observe some connective tissue fibers amid collagen fibers (Figure 8, F2D).

However, for the 90 days, the histological image of the implanted area was different from F1, where three zones were observed. Zone 1 was attributed to the material in the process of degradation/reabsorption, surrounded by a capsule. In Zone 2, the tissue appeared to have a higher level of healing. Finally, a third zone corresponded to the appearance of already-healed connective tissue (Figure 8, F2G).

When studying Zone 1 at higher magnification, an area demarcated by a capsule with inflammatory cells, including connective tissue of reticular appearance inside (Figure 8, F2H), was observed. At higher magnification (40×), some small fragments of the material in the middle of bundles of connective tissue of normal appearance were observed (Figure 8, F2I).

It is interesting to note that although the F1 and F2 formulations had differences in their compositions, with a decreased amount of PCL and additional TiO_2,_ using PLA as a support matrix with 70 wt.%, it influenced the degradation and resorption process of the material, with a faster healing process observable in 60 days, with a decrease in the inflammatory infiltrate and an almost total replacement of the material by connective tissue at 90 days. Additionally, adding TiO_2_ nanoparticles promoted decreased inflammatory infiltrate due to their biocompatible and biological properties [40].

The inflammatory infiltrates in both formulations were probably due to PLA-promoting phagocytic processes facilitated by inflammatory cells, thanks to the fragmentation of the primitive structure attributed to attacks by water on hydro-labile bonds, which led to unusual thermal, biocompatible, crystalline, and degradative properties [41]. On the other hand, PCL is also considered a biocompatible material; however, it has been reported that fibers obtained by electrospinning have slow resorption [42]. In this sense, it would explain how PCL influences material biodegradability concerning healing, since a decrease in the concentration in weight of PCL leads to a faster rate of healing observable within 60 days, with a reduction in the inflammatory infiltrates and an almost total replacement of the material by connective tissue at 90 days.

An interesting finding was the formation of connective tissue with a reticular arrangement within the capsule, as observed in formulation F2, Figure 8, F2H. This finding could be related to a faster degradation/reabsorption of the implanted material and an improvement in biocompatibility that would cause an increase in scar tissue turnover and maturation.

On the other hand, for formulation F3 with a composition of 20% PCL:70% PLA:10% OEO, a different scarring process was observed. At 30 days, the implantation area showed material surrounded by a fibrous capsule (Figure 8, F3A). However, unlike the other two formulations, the implanted material was largely reabsorbed and surrounded by connective tissue with a reticular appearance (Figure 8, F3B). In addition, Masson’s trichrome staining showed the presence of collagen type I fibers in the capsule and contact with the resorbing material (Figure 8, F3C). Finally, at 60 days after implantation, the images indicated the presence of these reticulated connective tissue structures surrounded by a fibrous capsule (Figure 8, F3D), with the presence of some inflammatory cells in the capsule (Figure 8, F3E), and collagen fibers in the capsule and the interior (Figure 8, F3F).

Formulation F3 presents in the scarring process features in common with those found for formulation F2, such as the presence of connective tissue in a reticular arrangement. However, while in the second formulation, it was only possible to observe this arrangement in the 90 days, in formulation F3 it was present from 30 days, with remnant material at 90 days. According to the composition, F3 decreased the PCL amount but added 10 wt.% of OEO without TiO_2_.

The decrease in the percentage of PCL caused an increase in degradability. The incorporation of OEO improved the biocompatibility favored by the volatility of its components causing micro holes as observed in the SEM, which influenced a faster degradation/reabsorption of the material and replacement by connective tissue, preserving the reticular arrangement followed in the 60 days of F2, as similarly occurred in previous works [21].

In formulation F4 with a composition of 17% PCL:70% PLA:3% TiO_2_: 10% OEO, at 30 days of implantation (Figure 8, F4A), some areas with the material in the process of degradation by cellular infiltration were observed, with a histological image very similar to that reported for the first two formulations at 30 days (Figure 8, F1A and F2A), with the presence of remaining material immersed in a connective tissue made up of type 1 collagen bundles (Figure 8, F4B and F4C).

At 60 days after implantation, the appearance of the implantation zone was similar to that reported for formulation F3, with connective tissue with a reticular formation surrounded by a capsule (Figure 8, F4D and F4E) and the presence of some multinucleated inflammatory cells. However, after 90 days, the reticulated structure was no longer observed, and the implantation zone comprised highly organized connective tissue with some membrane residues during degradation/reabsorption.

Therefore, the F4 formulation exhibited higher biodegradability at 90 days of implantation with the appearance of connective tissue due to the incorporation of TiO_2_ and OEO, which acted synergistically. Additionally, the dispersion of components within the polymeric matrix was evidenced, causing a thermal reinforcement of the material as observed in the TGA.

This study observed a capsule surrounding the material in the implantation zone and collagen type I fibers inside the capsule, separating portions of the material from inflammatory cells. The capsule presence is explained by the strong material fragment resistance, such as PLA, which may encourage the foreign body reaction despite being biocompatible [43].

A capsule surrounding the implanted material is a normal finding during biomaterial studies in animal models. During the healing process, the acute inflammation caused by the surgical lesion occurs with the implanted material [44]. Once the acute phase is overcome, the remaining material produces a chronic inflammatory phase that finishes once the situation is resolved [45].

During chronic inflammation, the foreign body reaction in which the material is encapsulated limits the damage and allows the regenerative process to continue. Within the capsule, inflammatory cells will be present, responsible for degrading and reabsorbing the material. Once this occurs, the capsule disappears [46].

## 4. Conclusions

In this work, we synthesized and characterized PCL/PLA/TiO_2_/OEO membranes, demonstrating high thermal stability and homogeneously distributed surfaces, resulting in increased dispersion of all the components exhibiting enhanced biocompatibility with the appearance of connective tissue. The TiO_2_ nanoparticle addition without OEO shows small islands of this nanomaterial on the PLA phase due to the incompatibility with the PCL. Therefore, these nanoparticles influenced the crystallographic behavior because they restrict the mobility of PLA chains, while the addition of OEO facilitates the mobility of these chains. On the other hand, In the FTIR spectrum, the characteristic bands for each component were evidenced. Additionally, those containing OEO exhibited carbonyl bands broadening and loss of the intensity of the C-O-C bond at 1756 and 1296cm^−1^, respectively, due to hydrogen bonding. Thermal analysis elucidated a thermal strengthening for the mixture compared to the pure components, probably due to PCL’s more thermally stable presence. Moreover, the TiO_2_ decreased degradation temperatures, while OEO addition increased those temperatures.

All samples presented an inflammatory process with a lymphatic infiltrate component 30 days after implantation, which transferred to a mild inflammatory process at 90 days. In addition, the F4 membrane with a composition of 17% PCL:70% PLA:3% TiO_2_:10% OEO showed the most significant signs of resorption with the presence of connective tissue, indicating that the incorporation of OEO promotes biocompatibility. At the same time, TiO_2_ is related to biodegradability, stimulating a more significant interaction with phagocytic cells.

## Figures and Tables

**Figure 1 polymers-15-00135-f001:**
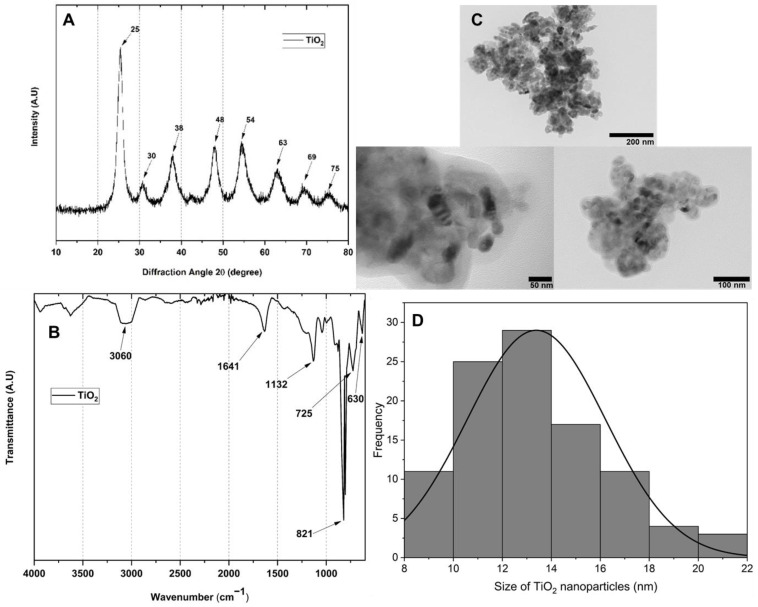
Characterization of TiO_2_ nanoparticles: (**A**) XRD (**B**) FTIR (**C**) TEM, and (**D**) histogram of the TEM nanoparticle.

**Figure 2 polymers-15-00135-f002:**
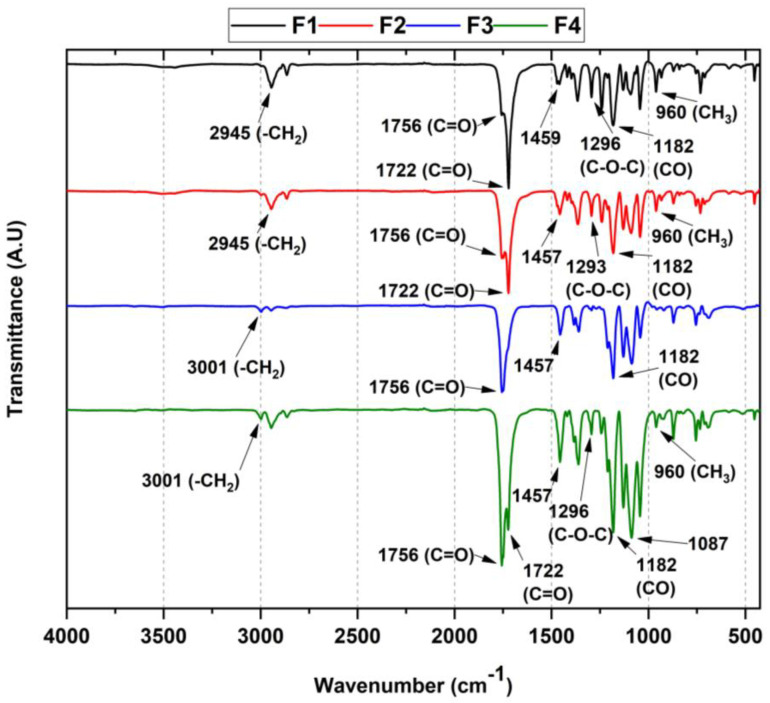
FT-IR of the four membranes (F1–F4 according to Table 1). F1, 30% PCL/70% PLA; F2, 27% PCL/70% PLA/3% TiO_2_; F3, 20% PCL:70% PLA:10% OEO; F4, 17% PCL:70% PLA:3% TiO_2_:10% OEO.

**Figure 3 polymers-15-00135-f003:**
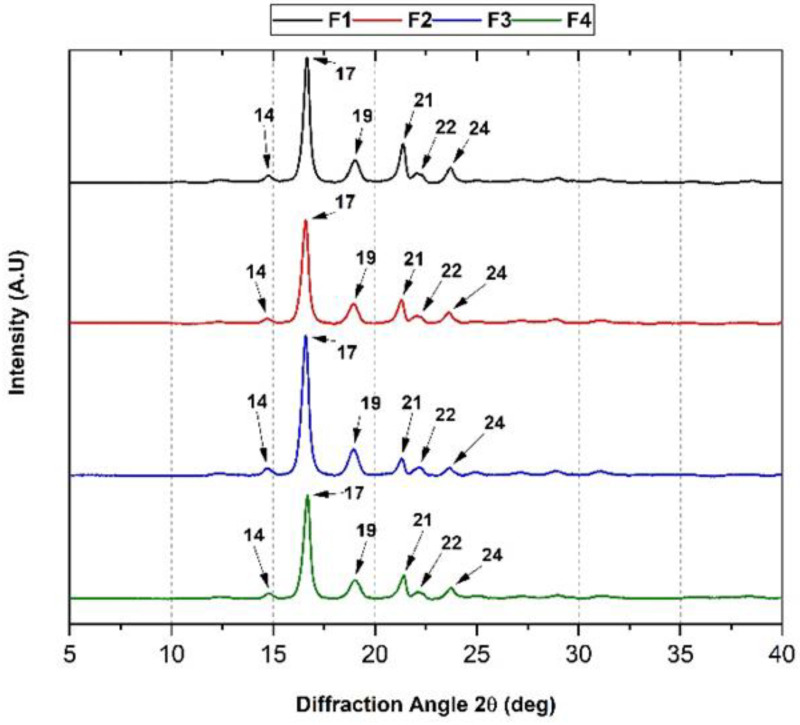
XRD analysis of the four membranes (F1–F4 according to Table 1). F1, 30% PCL/70% PLA; F2, 27% PCL/70% PLA/3% TiO_2_; F3, 20% PCL:70% PLA:10% OEO; F4, 17% PCL:70% PLA:3% TiO_2_:10% OEO.

**Figure 4 polymers-15-00135-f004:**
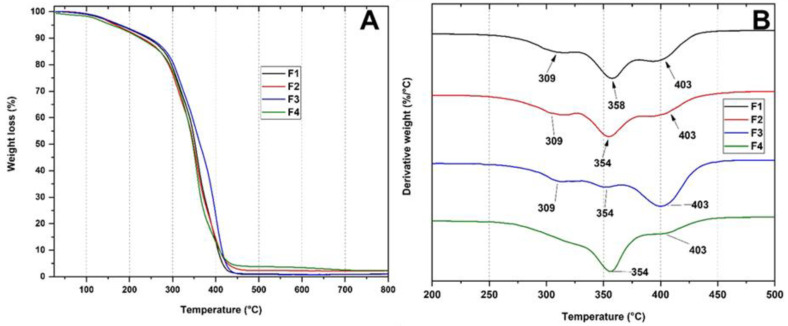
Thermal analysis of membranes PCL/PLA/TiO_2_/OEO. F1, 30% PCL/70% PLA; F2, 27% PCL/70% PLA/3% TiO_2_; F3, 20% PCL:70% PLA:10% OEO; F4, 17% PCL:70% PLA:3% TiO_2_:10% OEO. (**A**) TGA (**B**) DTGA.

**Figure 5 polymers-15-00135-f005:**
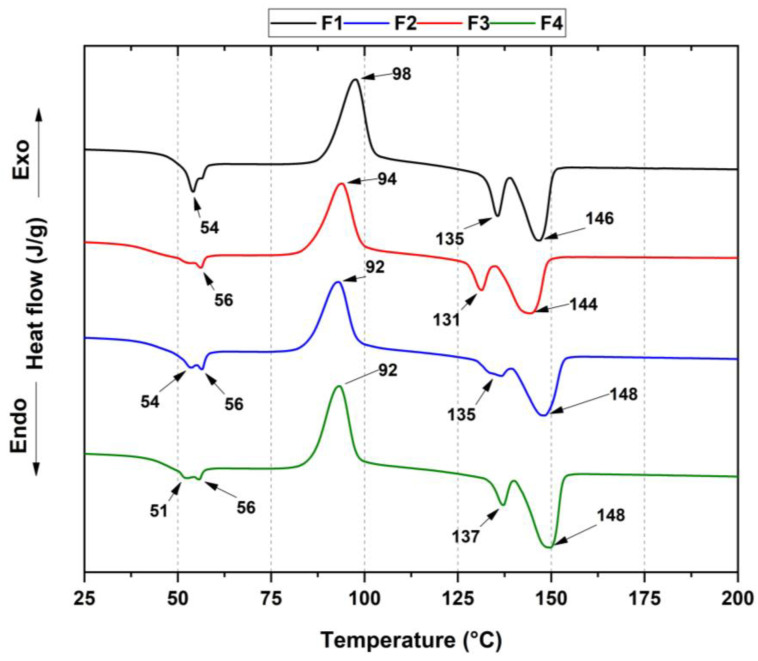
DSC thermogram for the membranes PCL/PLA/TiO_2_/OEO. F1, 30% PCL/70% PLA; F2, 27% PCL/70% PLA/3% TiO_2_; F3, 20% PCL:70% PLA:10% OEO; F4, 17% PCL:70% PLA:3% TiO_2_:10% OEO.

**Figure 6 polymers-15-00135-f006:**
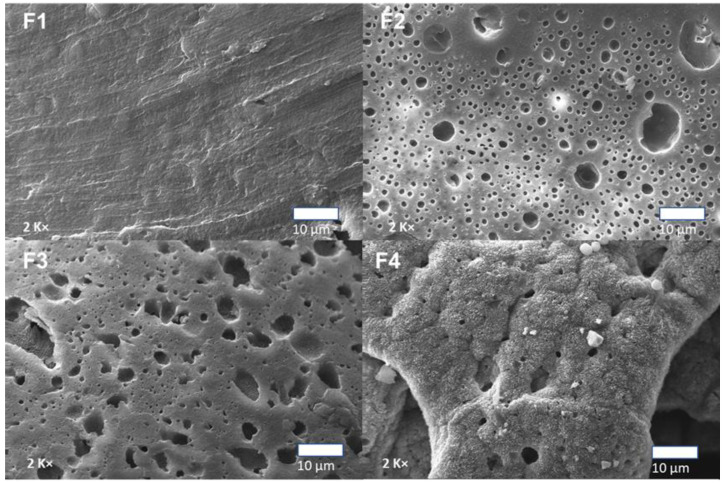
Scanning electron micrographs of membranes PCL/PLA/TiO_2_/OEO. F1, 30% PCL/70% PLA; F2, 27% PCL/70% PLA/3% TiO_2_; F3, 20% PCL:70% PLA:10% OEO; F4, 17% PCL:70% PLA:3% TiO_2_:10% OEO.

**Figure 7 polymers-15-00135-f007:**
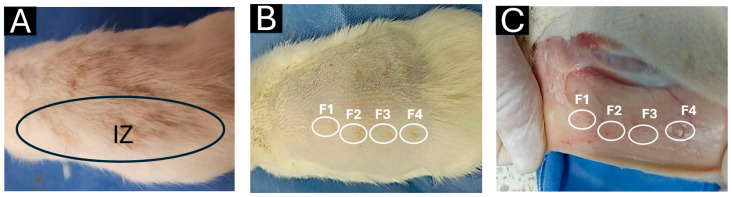
Subdermal dorsal implantation zone. (**A**) Dorsal area with abundant hair. (**B**) Trichotomy of the dorsal region. (**C**) Subdermal implantation area. Black oval: Implantation zone. White circles: Blocks implanted. F1–F4: Formulations 1, 2, 3, and 4. IZ: Implantation zone.

**Figure 8 polymers-15-00135-f008:**
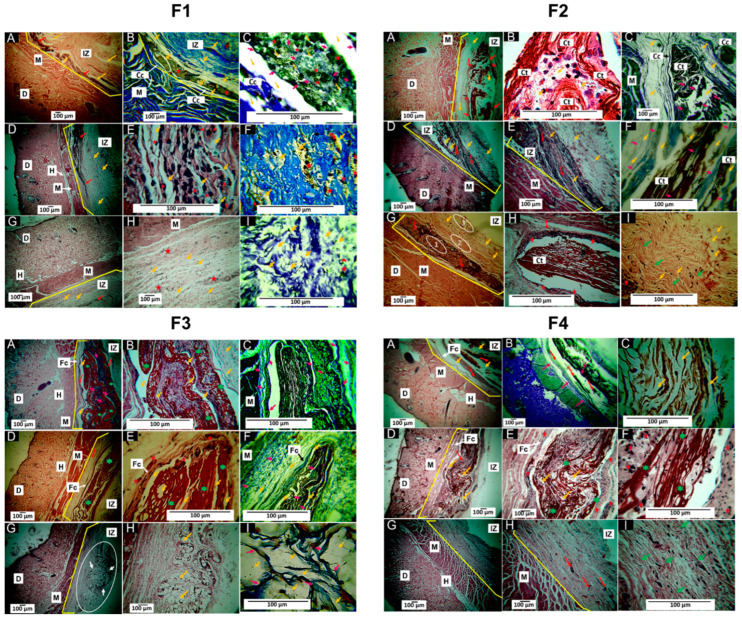
Histological analysis of F1, F2, F3, and F4 membranes. (**A**–**C**): 30-day implantations. (**D**–**F**): 60-day implantations. (**G**–**I**): Implantations at 90 days. Panel F1: (**A**): Image at 4× HE technique. (**B**): Image at 10× MT technique. (**C**): Image at 100× MT technique. (**D**): Image at 4× HE technique. (**E**): Image at 40× HE technique. (**F**): Image at 40× MT technique. (**G**): Image at 4× HE technique. (**H**): Image at 10× HE technique. (**I**): Image at 40× MT. Panel F2: (**A**): Image at 4× HE technique. (**B**): Image at 40× HE technique. (**C**): Image at 10× MT technique. (**D**): Image at 4× HE technique. (**E**): Image at 10× HE technique. (**F**): Image at 40× MT technique. (**G**): Image at 4× HE technique. (**H**): Image at 40× HE technique. (**I**): Image at 40× HE. Panel F3: (**A**): Image at 4× HE technique. (**B**): Image at 40× HE technique. (**C**): Image at 40× MT technique. (**D**): Image at 4× HE technique. (**E**): Image at 40× HE technique. (**F**): Image at 40× MT technique. (**G**): Image at 4× HE technique. (**H**): Image at 10× HE technique. (**I**): Image at 40× MT technique. Panel F4: (**A**): Image at 4× HE technique. (**B**): Image at 4× MT technique. (**C**): Image at 40× HE technique. (**D**): Image at 4× HE technique. (**E**): Image at 10× HE technique. (**F**): Image at 40× HE technique. (**G**): Image at 4× HE technique. (**H**): Image at 10× HE technique. (**I**): Image at 40× HE. D: Dermis. M: Muscle. IZ: Implantation zone. Cc: Connective tissue capsule. Ct: Connective tissue. Fc: Fibrous capsule. H: hypodermis. Red arrow: area with resorbing material. Yellow arrow: Material. Pink arrows: type I collagen fibers. Green arrows: Connective tissue. Red stars: Inflammatory cells. Green stars: Connective tissue. Oval 1: Area with the material in the process of resorption. Oval 2: Zone with less inflammatory activity. Oval 3: Zone with the formation of connective scar tissue. White oval: Histological interest zone. White arrows: material in the process of degradation/resorption. MT: Masson’s trichrome stain. HE: Hematoxylin–Eosin stain.

**Table 1 polymers-15-00135-t001:** Formulations used for forming PCL/PLA/TiO_2_/OEO membranes (wt.%).

Components	F1	F2	F3	F4
PCL (%)	30	27	20	17
PLA (%)	70	70	70	70
TiO_2_ (%)	0	3	0	3
OEO (%)	0	0	10	10

**Table 2 polymers-15-00135-t002:** Thermal properties of membranes PCL/PLA/TiO_2_/OEO. F1, 30% PCL:70% PLA; F2, 27% PCL/70% PLA/3% TiO_2_; F3, 20% PCL:70% PLA:10% OEO; F4, 17% PCL:70% PLA:3% TiO_2_:10% OEO.

	T_g_ (°C)	T_cc_ (°C)	T_m1_ (°C)	T_m2_ (°C)	T_m3_ (°C)	Xc PLA (%)
F1	54	98	135	146	54	5.2
F2	56	94	131	144	56	1.8
F3	54	92	135	148	56	4.4
F4	51	92	137	148	56	0.6

## Data Availability

Data available under request to the corresponding author.

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
