# Peer review of "Synthesis, Characterization, and Optimization Studies of Polycaprolactone/Polylactic Acid/Titanium Dioxide Nanoparticle/Orange Essential Oil Membranes for Biomedical Applications"

_polymers, 2022, doi:10.3390/polym15010135_

Round 1

Reviewer 1 Report

In this article, the authors synthesize PCL/PLA/TiO2/OEO composite membranes and target them for application in artificial scaffolds. These composite membranes exhibit great thermal stability and biocompatibility due to the incorporation of OEO and TiO2. the article has a clear idea, clear framework, and logical language; thus I enthusiastically support acceptance of this work after addressing the following issues.

(1)    Please refine the main innovations of this paper and the main contributions to the artificial scaffolds

(2)    The abstract is not written succinctly enough; the main idea is not expressed through the abstract. And "F4" is inappropriate with no definition.

(3)    In lines 259-260, the peak of 438cm-1 does not exist without direct evidence. In Figure 2, the minimum scale is 500cm-1

(4)    The clarity of the picture is not high, for example, the words in Figure 7 are not legible.

(5)    We recommend that the articles published in Advanced Powder materials, Advanced Fiber Materials, and Chinese Chemical Letters journals be cited.

Author Response

We are deeply thankful for the valuable comments that support improving the manuscript's quality. The corrections are presented below point by point in red for easy comprehension. All the answers are given in the attached version. 

Reviewer 1

  • Please refine the main innovations of this paper and the main contributions to the artificial scaffolds

R// Thank you very much for the suggestion. The new abstract introduced the respective innovations to show why this composition of several elements in a membrane and analyze their response in vivo. Additionally, the introduction specifies why we introduced titanium dioxide nanoparticles and the orange essential oil.

Abstract

The development of scaffolds for cell regeneration has increased because they must have adequate biocompatibility and mechanical properties to be applied in tissue engineering. In this sense, incorporating nanofillers or essential oils has allowed the realization of new architectures with the possibility of promoting cell proliferation and regeneration of new tissue. Line 22-36

Introduction

However, despite the study in different fields of PLA nanocomposites with nanoparticles, there is a need to improve the stability and biocompatibility under physiological conditions to prolong the durability of nanocomposites with physical mixtures containing a PCL/PLA polymeric matrix as well as an orange essential oil. There is still a great need to determine whether blending nanofillers with other components improve biocompatibility, especially in vivo studies. (Lines 94-99).

  • The abstract is not written succinctly enough; the main idea is not expressed through the And "F4" is inappropriate with no definition.

R// Thank you very much for the suggestion. We chose to reduce the content in the new summary by focusing on the main observations. In addition, to avoid talking about a specific formulation, we spoke in a general way emphasizing tissue regeneration by the incorporation of the materials which together promote the decrease of the inflammatory infiltrate as well as the appearance of connective tissue.

Abstract: The development of scaffolds for cell regeneration has increased because they must have adequate biocompatibility and mechanical properties to be applied in tissue engineering. In this sense, incorporating nanofillers or essential oils has allowed new architectures to promote cell proliferation and regeneration of new tissue. With this goal, we prepared four membranes based on polylactic acid (PLA), polycaprolactone (PCL), titanium dioxide nanoparticles (TiO2-NPs), and orange essential oil (OEO) by the drop-casting method. The preparation of TiO2-NPs followed the sol-gel process with spherical morphology and an average size of 13.39 nm ± 2.28 nm. The results show how the TiO2-NP properties predominate over the crystallization processes, reflected in the decreasing crystallinity percentage from 5.2 to 0.6% in the membranes.

On the other hand, when OEO and TiO2-NPs are introduced into a membrane, they act synergistically due to the inclusion of highly conjugated thermostable molecules and the thermal properties of TiO2-NPs. Finally, incorporating OEO and TiO2-NPs promotes tissue regeneration due to the decrease of inflammatory infiltrate and the appearance of connective tissue. These results demonstrate the great potential for biomedical applications of the membranes prepared.    

(3)    In lines 259-260, the peak of 438cm-1 does not exist without direct evidence. In Figure 2, the minimum scale is 500cm-1

R// Thank you very much for the observation. We increased the range of the spectrum. However, in the new range between 425-4000 cm-1, the band corresponding to TiO2-NPs is absent due to the small amount of this component in the membranes; in addition, the band could be confused with the noise of the equipment. However, as you can see from the text, we were never assured that the band was present. On the contrary, we are saying that: "the characteristic band at 438 cm-1 was absent, probably due to the tiny amount by weight for each formulation." (Lines 265-267).

(4)    The clarity of the picture is not high. For example, the words in Figure 7 are not legible.

R// Thank you very much for the observation. We increase the size of the graphic so that the letters within the image are visible. However, one reviewer from a previously published work suggested we put together all the histology pictures, which is why they are presented like that.

(5)    We recommend that the articles be published in Advanced Powder materials, Advanced Fiber.

R// Thank you very much for the recommendation. The following references were added:

  1. Haroosh, H.; Chaudhary, D.; Dong, Y.; Hawkins, B. Electrospun PLA: PCL/halloysite nanotube nanocomposites fibers for drug delivery. Process. Fabr. Adv. Mater. XIX 2011, 847–858.
  2. Wang, L.; Zhang, F.; Liu, Y.; Leng, J. Shape memory polymer fibers: materials, structures, and applications. Adv. Fiber Mater. 2022, 4, 5–23.
  3. Wei, Y.; Zhu, J.; Gan, Y.; Cheng, G. Titanium glycolate-derived TiO2 nanomaterials: Synthesis and applications. Adv. Powder Technol. 2018, 29, 2289–2311.

Reviewer 2 Report

This manuscript studies the structural, thermal and biocompatibility performances of polycaprolactone/ polylactic acid/ TiO2 nanoparticles/ orange essential oil membranes. The subject is interesting and the manuscript has been well written. However, it needs a revision before the final decision.

1. The title needs revision. It is recommended not to use the abbreviations in the title. Also, the title needs more information about the methodology.

2. You have used F4 sample in Abstract; however, the reader does not know what sample F4 is. Please resolve this issue.

3. PLA has been used three times in lines 53-56; however, its full name has been introduced in line 60. Also, PCL has been used in line 57 and line 58; however, it has been introduced in line 60! Please carefully check the abbreviations. They should be defined in their first presentation. PCL has been introduced again in 102. PCL and PLA have been introduced again in line 239!

4. Please carefully check all abbreviations throughout the manuscript. OEO has been introduced in full name dozens of time, for instance, in the title, in line 24, in line 77, in line 132, in line 133, in line 229 and etc.

5. Please change the heading of the section three to “Results and discussion”.

Author Response

We are deeply thankful for the valuable comments that support improving the manuscript's quality. The corrections are presented below point by point in red for easy comprehension. All the answers are given in the attached version. 

Reviewer 2

  1. The title needs revision. It is recommended not to use abbreviations in the title. Also, the title needs more information about the methodology.

R// Thank you very much for the recommendation. We changed the title, avoiding abbreviations as follows:  

Synthesis, characterization, and optimization studies of polycaprolactone/polylactic acid/titanium dioxide nanoparticles/orange essential oil membranes for biomedical applications.

  1. You have used F4 sample in Abstract; however, the reader does not know what sample F4 is. Please resolve this issue.

R// Thank you very much for your comments. The abstract was rewritten based on a reviewer's recommendation. Now, the F4 term is not present in the new version.

  1. PLA has been used three times in lines 53-56; however, its full name has been introduced in line 60. Also, PCL has been used in line 57 and line 58; however, it has been introduced in line 60! Please carefully check the abbreviations. They should be defined in their first presentation. PCL has been introduced again in 102. PCL and PLA have been introduced again in line 239.

R// Thank you very much for the observation. All the polylactic acid and polycaprolactone names were replaced by their acronyms, PLA and PCL.

  1. Please carefully check all abbreviations throughout the manuscript. OEO has been introduced in full name dozens of times, for instance, in the title, in line 24, in line 77, in line 132, in line 133, in line 229 etc.

R// Thank you very much for the observation. The OEO abbreviation was explained in line 25. From there, the acronym was introduced. 

  1. Please change the heading of section three to "Results and discussion".

R// Thank you very much for the recommendation. Section three has already been changed to results and discussion.

Reviewer 3 Report

In this contribution, the authors prepared the PCL/PLA/TiO2/OEO membranes, and systematically studied their crystallographic and morphological behaviors, as well as their biocompatibility. This topic is inspiring to the readership of Polymers. However, several controversial discussions necessitate the following questions/comments to be addressed before further decisions.

1. In FTIR spectra in Figure 2, the peaks between 2945-3001 cm-1 are assigned to the symmetric and asymmetric strain bands of the CH bonds (Line 250). But why does only F3, not also F4, show the uniquely high intensity at 3001 cm-1?

2. Which component do the pores result from in Figure 5? In Line 290, the formation of pores is attributed to the evaporation of volatile OEO. However, why does F2 comprising no OEO also show a porous surface? In Line 30, the porous morphologies are attributed to TiO2, but why is F3 containing no TiO2 also porous?

3. The major component of OEO, limonene, has a boiling point of 176 °C. But why do F3 and F4 in Figure 4A show no weight loss associated with the removal of limonene? In addition, how to distinguish the weight loss due to the thermal degradation of PLA and PCL?

4. Figure 4C, the DSC results mentioned in Line 325, is not included.

5. The TiO2 nanoparticles restrict the mobility of PLA chains, so F2 and F4 show lower Xc,PLA than F1 and F3, respectively (Line 535 and Table 2). However, if the OEO facilitates the mobility of PLA chains (Line 535), why do F3 and F4 show lower Xc,PLA than F1 and F2, respectively?

6. Other amendments:

Line 28 (also in SI), the presence of OEO decreases, instead of increasing, the Tg of PLA.

Line 221 introduces 8 peaks (48° twice), but only 6 planes.

Line 271, XRD instead of DRX.

Author Response

We are deeply thankful for all the valuable comments that helped us to improve the manuscript's quality. The corrections are presented below point by point in red for easy comprehension.

Reviewer 3

  1. In FTIR spectra in Figure 2, the peaks between 2945-3001 cm-1are assigned to the symmetric and asymmetric strain bands of the CH bonds (Line 250). But why does only F3, not also F4, show the uniquely high intensity at 3001 cm-1?

R// Thank you very much for the observation. In the FTIR spectrum, the correction in the band signaling was made. Additionally, it must be considered that there is a mixture of components where each one contributes the characteristic bands depending on the chemical environment as well as the vibrational frequencies, i.e., that probably more than one band corresponding to the CH2 groups is present due to the electronic environments that the components have.

  1. Which component do the pores result from in Figure 5? In Line 290, the formation of pores is attributed to the evaporation of volatile OEO. However, why does F2 comprising no OEO also show a porous surface? In Line 30, the porous morphologies are attributed to TiO2, but why is F3 containing no TiO2also porous?

R// Thank you very much for your question. Nanoparticles, especially TiO2 or ZnO, generally produce porous surfaces because they promote the degradation of the polymeric matrix. Moreover, it has been observed that this morphology promotes cellular proliferation and decreases thermal properties, as in the young model. On the other hand, essential oils are made up of many organic components with boiling points below 40 °C; therefore, when membranes are subjected to a preheated oven at this temperature, these components will volatilize, generating porosity in the membrane.

  1. The major component of OEO, limonene, has a boiling point of 176 °C. But why do F3 and F4 in Figure 4A show no weight loss associated with the removal of limonene? In addition, how to distinguish the weight loss due to the thermal degradation of PLA and PCL?

R// Thank you very much for your question. Indeed, in the TGA, the thermal degradation associated with the release of limonene is not shown, probably because this natural component is integrated into the polymeric matrix, granting an increase in the material's thermal properties, as shown for the F3 formulation. Previous works have demonstrated the same activity where the thermal degradation corresponding to the natural product is not shown. Still, it is seen as a reinforcement of the thermal properties due to the high conjugation and thermal stability [1,2].

  1. Figure 4C, the DSC results mentioned in Line 325, is not included.

R// Thank you very much for the observation. When the different thermograms were inserted, the DSC thermogram was omitted, so it is added below and mentioned in the text (Line 330):

Figure 5. DSC thermogram for the membranes PCL/PLA/TiO2/OEO. F1, 30%PCL/70%PLA; F2, 27%PCL/70%PLA/3%TiO2; F3, 20%PCL:70%PLA:10%OEO; F4, 17%PCL:70%PLA:3%TiO2:10%OEO.

  1. The TiO2nanoparticles restrict the mobility of PLA chains, so F2 and F4 show lower Xc,PLAthan F1 and F3, respectively (Line 535 and Table 2). However, if the OEO facilitates the mobility of PLA chains (Line 535), why do F3 and F4 show lower Xc,PLA than F1 and F2, respectively?

R// Thank you very much for the question. As seen in table 2, nanoparticles decrease the crystallization percentage because the nucleation centers are unavailable due to the low mobility of the alkyl chains that PLA possesses. Therefore, the crystal growth is impeded, which causes its crystallization percentage to decrease. On the other hand, as observed in F3, the crystallization percentage increases because the nucleation process is favored due to the low viscosity of the essential oil, which increases the mobility of the alkyl chains; in addition, it was observed that the effect of the titanium dioxide nanoparticles predominates over the impact that the orange essential oil can provide [3].

  1. Other amendments:

Line 28 (also in SI), the presence of OEO decreases, instead of increasing, the Tg of PLA.

R// Thank you very much for the observation. We rewrote the abstract as one of the reviewers requested. The new abstract does not have this information, avoiding confusion.

Line 221 introduces 8 peaks (48° twice), but only 6 planes.

R// Thank you very much for the observation. The correction of the crystallographic drawings has already been made as follows (Lines 227-228): 25°, 38°, 48°, 54°, 63°, 69°, and 75° corresponding to the (101), (004), (200), (211), (105), (204) and (220) planes for TiO2

Line 271, XRD instead of DRX.

R// Thank you very much for the observation. The correction of the acronym has been made.

XRD of the PLA/PCL/TiO2/OEO membranes (Line 277).

References 

  1. Wu, Y.; Qin, Y.; Yuan, M.; Li, L.; Chen, H.; Cao, J.; Yang, J. Characterization of an antimicrobial poly (lactic acid) film prepared with poly (ε‐caprolactone) and thymol for active packaging. Polym. Adv. Technol. 2014, 25, 948–954.
  2. Fortunati, E.; Luzi, F.; Puglia, D.; Dominici, F.; Santulli, C.; Kenny, J.M.; Torre, L. Investigation of thermo-mechanical, chemical and degradative properties of PLA-limonene films reinforced with cellulose nanocrystals extracted from Phormium tenax leaves. Eur. Polym. J. 2014, 56, 77–91.
  3. Yahyaoui, M.; Gordobil, O.; Herrera Díaz, R.; Abderrabba, M.; Labidi, J. Development of novel antimicrobial films based on poly(lactic acid) and essential oils. React. Funct. Polym. 2016, 109, 1–8, doi:10.1016/j.reactfunctpolym.2016.09.001.

Round 2

Reviewer 3 Report

Thank you for the thorough point-to-point reply. All my questions and comments have been addressed.